# THE ROLE OF EMBEDDING COMPLEXITY IN DOMAIN-INVARIANT REPRESENTATIONS

## ABSTRACT

Unsupervised domain adaptation aims to generalize the hypothesis trained in a source domain to an unlabeled target domain. One popular approach to this problem is to learn domain-invariant embeddings for both domains. In this work, we study, theoretically and empirically, the effect of the embedding complexity on generalization to the target domain. In particular, this complexity affects an upper bound on the target risk; this is reflected in experiments, too. Next, we specify our theoretical framework to multilayer neural networks. As a result, we develop a strategy that mitigates sensitivity to the embedding complexity, and empirically achieves performance on par with or better than the best layer-dependent complexity tradeoff.

## 1 INTRODUCTION

Domain adaptation is critical in many applications where collecting large-scale supervised data is prohibitively expensive or intractable, or where conditions at prediction time can change. For instance, self-driving cars must be robust to different weather, change of landscape and traffic. In such cases, the model learned from limited source data should ideally generalize to different target domains. Specifically, unsupervised domain adaptation aims to transfer knowledge learned from a labeled source domain to similar but completely unlabeled target domains.

One popular approach to unsupervised domain adaptation is to learn domain-invariant representations (Ben-David et al., 2007; Long et al., 2015; Ganin et al., 2016), by minimizing a divergence between the representations of source and target domains. The prediction function is learned on these "aligned" representations with the aim of making it domain-independent. A series of theoretical works justifies this idea (Ben-David et al., 2007; Mansour et al., 2009; Ben-David et al., 2010; Cortes & Mohri, 2011).

Despite the empirical success of domain-invariant representations, exactly matching the representations of source and target distribution can sometimes fail to achieve domain adaptation. For example, Wu et al. (2019) show that exact matching may increase target error if label distributions are different between source and target domain, and propose a new divergence metric to overcome this limitation. Zhao et al. (2019) establish lower and upper bounds on the risk when label distributions between source and target domains differ. Johansson et al. (2019) point out the information lost in non-invertible embeddings, and propose different generalization bounds based on the overlap of the supports of source and target distribution.

In contrast to previous analyses that focus on changes in the label distributions or joint support, we study the effect of embedding complexity. In particular, we show a general bound on the target risk that reflects a tradeoff between embedding complexity and the divergence of source and target domains. A too powerful class of embeddings can result in overfitting the source data and the matching of source and target distributions, resulting in arbitrarily high target risk. Hence, a restriction is needed. We observe that indeed, without appropriately constraining the embedding complexity, the performance of state-of-the-art methods such as domain-adversarial neural networks (Ganin et al., 2016) can deteriorate significantly.

Next, we tailor the bound to multilayer neural networks. In a realistic scenario, one may have a total depth budget and divide the network into an encoder (embedding) and predictor by aligning the representations of source and target in a chosen layer, which defines the division. In this case,

a more complex encoder necessarily implies a weaker predictor, and vice versa. This tradeoff is reflected in the bound and, we see that, in practice, there is an "optimal" division.

To better optimize the tradeoff between encoder and predictor without having to tune the division, we propose to optimize the tradeoffs in all layers jointly via a simple yet effective objective that can easily be combined with most current approaches for learning domain-invariant representations. Implicitly, this objective restricts the more powerful deeper encoders by encouraging a simultaneous alignment across layers. In practice, the resulting algorithm achieves performance on par with or better than standard domain-invariant representations, without tuning of the division.

Empirically, we examine our theory and learning algorithms on sentiment analysis (Amazon review dataset), digit classification (MNIST, MNIST-M, SVHN) and general object classification (Office-31). In short, this work makes the following contributions:

- General upper bounds on target error that capture the effect of embedding complexity when learning domain-invariant representations;
- Fine-grained analysis for multilayer neural networks, and a new objective with implicit regularization that stabilizes and improves performance;
- Empirical validation of the analyzed tradeoffs and proposed algorithm on several datasets.

## 2 UNSUPERVISED DOMAIN ADAPTATION

For simplicity of exposition, we consider binary classification with input space $\mathcal{X} \subseteq \mathbb{R}^n$ and output space $\mathcal{Y} = \{0, 1\}$. Define $\mathcal{H}$ to be the hypothesis class from $\mathcal{X}$ to $\mathcal{Y}$. The learning algorithm obtains two datasets: labeled source data $\mathcal{X}_S$ from distribution $p_S$, and unlabeled target data $\mathcal{X}_T$ from distribution $p_T$. We will use $p_S$ and $p_T$ to denote the joint distribution on data and labels $X, Y$ and the marginals, i.e., $p_S(X)$ and $p_S(Y)$. Unsupervised domain adaptation seeks a hypothesis $h \in \mathcal{H}$ that minimizes the risk in the target domain measured by a loss function $\ell$ (here, zero-one loss):

$$R_T(h) = \mathbb{E}_{x,y \sim p_T}[\ell(h(x), y)]. \tag{1}$$

We will not assume common support in source and target domain, in line with standard benchmarks for domain adaptation such as adapting from MNIST to MNIST-M.

### 2.1 DOMAIN-INVARIANT REPRESENTATIONS

A common approach to domain adaptation is to learn a joint embedding of source and target data (Ganin et al., 2016; Tzeng et al., 2017). The idea is that aligning source and target distributions in a latent space $\mathcal{Z}$ results in a domain-invariant representations, and hence a subsequent classifier $f$ from the embedding to $\mathcal{Y}$ will generalize from source to target. Formally, this results in the following objective function on the hypothesis $h = fg := f \circ g$, where $\mathcal{G}$ is the class of embedding functions $g$ to $\mathcal{Z}$, and we minimize a divergence $d$ between the distributions $p_S^g(Z) = p_S(g(X)), p_T^g(Z) = p_T(g(X))$ of source and target after mapping to $\mathcal{Z}$:

$$\min_{f \in \mathcal{F}, g \in \mathcal{G}} R_S(fg) + \alpha d(p_S^g(Z), p_T^g(Z)). \tag{2}$$

The divergence $d$ could be, e.g., the Jensen-Shannon (Ganin et al., 2016) or Wasserstein distance (Shen et al., 2017).

### 2.2 UPPER BOUNDS ON THE TARGET RISK

Ben-David et al. (2007) introduced the $\mathcal{H}\Delta\mathcal{H}$-divergence to bound the worst-case loss from extrapolating between domains. Let $R_D(h, h') = \mathbb{E}_{x \sim D}[\ell(h(x), h'(x))]$ be the expected disagreement between two hypotheses. The $\mathcal{H}\Delta\mathcal{H}$-divergence measures whether there is any pair of hypotheses whose disagreement (risk) differs a lot between source and target distribution.

**Definition 1.** ($\mathcal{H}\Delta\mathcal{H}$-divergence) *Given two domain distributions $p_S$ and $p_T$ over $\mathcal{X}$, and a hypothesis class $\mathcal{H}$, the $\mathcal{H}\Delta\mathcal{H}$-divergence between $p_S$ and $p_T$ is*

$$d_{\mathcal{H}\Delta\mathcal{H}}(p_S, p_T) = \sup_{h,h' \in \mathcal{H}} |R_S(h, h') - R_T(h, h')|.$$

The $\mathcal{H}\Delta\mathcal{H}$-divergence is determined by the discrepancy between source and target distribution and the complexity ofthe hypothesis class $\mathcal{H}$. For a hypothesis class $\mathcal{H} : \mathcal{X} \to \{0, 1\}$, the disagreement between two hypotheses is equivalent to the exclusive or function. Hence, one can interpret the $\mathcal{H}\Delta\mathcal{H}$-divergence as finding a classifier in function space $\mathcal{H}\Delta\mathcal{H} = \mathcal{H} \oplus \mathcal{H}$ which attempts to maximally separate one domain from the other (Ben-David et al., 2010). A restrictive hypothesis space may result in small $\mathcal{H}\Delta\mathcal{H}$-divergence even if the source and target domain do not share common support. This divergence allows us to bound the risk on the target domain:

**Theorem 2.** (Ben-David et al., 2010) *For all hypotheses $h \in \mathcal{H}$, the target risk is bounded as*

$$R_T(h) \leq R_S(h) + d_{\mathcal{H}\Delta\mathcal{H}}(p_S, p_T) + \lambda_{\mathcal{H}},$$

*where $\lambda_{\mathcal{H}}$ is the best joint risk*

$$\lambda_{\mathcal{H}} := \inf_{h' \in \mathcal{H}} [R_S(h') + R_T(h')]$$

Similar results exist for continuous labels (Cortes & Mohri, 2011; Mansour et al., 2009).

Theorem 2 is an influential theoretical result in unsupervised domain adaptation, and motivated work on domain invariant representations. For example, recent work (Ganin et al. (2016); Johansson et al. (2019)) applied Theorem 2 to the hypothesis space $\mathcal{F}$ that maps the representation space $\mathcal{Z}$ induced by an encoder $g$ to the output space:

$$R_T(fg) \leq R_S(fg) + d_{\mathcal{F}\Delta\mathcal{F}}(p_S^g(Z), p_T^g(Z)) + \lambda_{\mathcal{F}}(g) \tag{3}$$

where $\lambda_{\mathcal{F}}(g)$ is the best hypothesis risk with fixed $g$, i.e., $\lambda_{\mathcal{F}}(g) := \inf_{f' \in \mathcal{F}} [R_S(f'g) + R_T(f'g)]$. The $\mathcal{F}\Delta\mathcal{F}$ divergence implicitly depends on the fixed $g$ and can be small if $g$ provides a suitable representation. However, if $g$ induces a wrong alignment, then the best hypothesis risk $\lambda_{\mathcal{F}}(g)$ is large with any function class $\mathcal{F}$. The following example will illustrate such a situation, motivating to explicitly take the class of embeddings into account when bounding the target risk.

## 3 INFLUENCE OF THE EMBEDDING COMPLEXITY

We begin with an illustrative toy example. Figure 1 shows a binary classification problem in 2D with disjoint support and a slight shift in the label distributions from source to target: $p_S(y = 1) = p_T(y = 1) + 2\epsilon$. Assume the representation space $\mathcal{Z}$ is one dimensional, so the embedding $g$ is a function from 2D to 1D. If we allow arbitrary, nonlinear embeddings, then, for instance, the embedding in Figure 1(b), together with an optimal predictor, achieves zero source loss and a zero divergence which is optimal according to the objective in equation (2). But the target risk of this combination of embedding and predictor is maximal: $R_T(fg) = 1$.

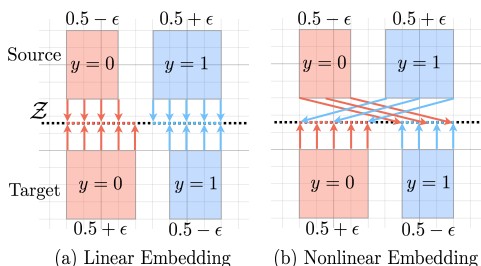

(a) Linear Embedding    (b) Nonlinear Embedding

Figure 1: Illustrative example in 2D. The 1D representation space is illustrated as a dotted line, and arrows indicate the embedding from 2D to 1D. (a) Optimal embedding when $\mathcal{G}$ is the class of linear functions. (b) Optimal embedding with a complex nonlinear function class: zero source error and divergence loss, but the embedding destroys label consistency and leads to maximal target risk.

If we restrict the class $\mathcal{G}$ of embeddings to linear maps $g(x) = \mathbf{W}x$ where $\mathbf{W} \in \mathbb{R}^{1 \times 2}$, then the embeddings that are optimal with respect to the objective (2) are of the form $\mathbf{W} = [a, 0]$. Together with an optimal source classifier $f$, they achieve a non-zero value of $2\epsilon$ for objective (2) due to the shift in class distributions. However, these embeddings retain label correspondences and can thus minimize target risk.

This example illustrates that a too rich class of embeddings can "overfit" the alignment, and hence lead to arbitrarily bad solutions. Hence, the complexity of the encoder class plays an important role in learning domain invariant representations.

### 3.1 BOUNDS FOR DOMAIN-INVARIANT REPRESENTATIONS

Motivated by the above example, we next expose how the bound on the target risk depends on the complexity of the embedding class. To do so, we apply Theorem 2 to the hypothesis $h = fg$:

$$R_T(fg) \leq R_S(fg) + d_{\mathcal{FG}\Delta\mathcal{FG}}(p_S, p_T) + \lambda_{\mathcal{FG}}. \tag{4}$$

This bound differs in two ways from the previous bound (equation (3)), which was based only on $\mathcal{F}$: the best in-class joint risk now minimizes over both $\mathcal{F}$ and $\mathcal{G}$, i.e.,

$$\lambda_{\mathcal{FG}} \coloneqq \inf_{f \in \mathcal{F}, g \in \mathcal{G}} [R_S(fg) + R_T(fg)], \tag{5}$$

which is smaller than $\lambda_{\mathcal{F}}(g)$ and reflects the fact that we are learning both $f$ and $g$. In return, the divergence term $d_{\mathcal{FG}\Delta\mathcal{FG}}(p_S, p_T)$ becomes larger than the one in equation (3). To better understand these tradeoffs, we will reformulate bound (4) to be more interpretable. To this end, we define a version of the $\mathcal{H}\Delta\mathcal{H}$-divergence that explicitly measures variation of the embeddings in $\mathcal{G}$:

**Definition 3.** ($\mathcal{F}_{\mathcal{G}\Delta\mathcal{G}}$-divergence) *For two domain distributions $p_S$ and $p_T$ over $\mathcal{X}$, an encoder class $\mathcal{G}$, and predictor class $\mathcal{F}$, the $\mathcal{F}_{\mathcal{G}\Delta\mathcal{G}}$-divergence between $p_S$ and $p_T$ is*

$$d_{\mathcal{F}_{\mathcal{G}\Delta\mathcal{G}}}(p_S, p_T) = \sup_{f \in \mathcal{F}; \, g, g' \in \mathcal{G}} |R_S(fg, fg') - R_T(fg, fg')|.$$

Importantly, the $\mathcal{F}_{\mathcal{G}\Delta\mathcal{G}}$-divergence is smaller than the $\mathcal{FG}\Delta\mathcal{FG}$-divergence, since the two hypotheses in the supremum, $fg$ and $fg'$, share the same predictor $f$.

**Theorem 4.** *For all $f \in \mathcal{F}$ and $g \in \mathcal{G}$,*

$$R_T(fg) \leq R_S(fg) + \underbrace{d_{\mathcal{F}\Delta\mathcal{F}}(p_S^g(Z), p_T^g(Z))}_{\text{Latent Divergence}} + \underbrace{d_{\mathcal{F}_{\mathcal{G}\Delta\mathcal{G}}}(p_S, p_T)}_{\text{Embedding Complexity}} + \lambda_{\mathcal{FG}}(g). \tag{6}$$

*where $\lambda_{\mathcal{FG}}(g)$ is the best in-class joint risk defined as*

$$\lambda_{\mathcal{FG}}(g) = \inf_{f' \in \mathcal{F}, g' \in \mathcal{G}} 2R_S(f'g) + R_S(f'g') + R_T(f'g').$$

We prove all theoretical results in the Appendix. This target generalization bound is small if (C1) the source risk is small, (C2) the latent divergence is small (because the domains are well-aligned and/or $\mathcal{F}$ is restricted), (C3) the complexity of $\mathcal{G}$ is restricted to avoid overfitting of alignments, and (C4) good source and target risk is in general achievable with $\mathcal{F}$ and $\mathcal{G}$.

**Comparison to Previous Bounds.** The last two terms in Theorem 2 express a similar complexity tradeoff, but with respect to the overall hypothesis class $\mathcal{H}$, which here combines encoder and predictor. Directly applying Theorem 2 to the composition $\mathcal{H} = \mathcal{FG}$ (equation (4)) treats both jointly and does not make the role of the embedding as explicit as Theorem 4. The recent bound (3) assumes a fixed embedding $g$ and focuses on the predictor class $\mathcal{F}$. As a result, it captures embedding complexity even less explicitly: the first two terms in bound (3) and Theorem 4 are the same. The last term in (3), $\lambda_{\mathcal{F}}(g)$, contains the target risk with the given $g$. Hence, bound (3) replaces (C3) and (C4) above by saying $\mathcal{F}$ and the specific $g$ (which is much harder to control since in practice it is also optimized) can achieve good source and target risk. In contrast, Theorem 4 states an explicit complexity penalty on the variability of the embeddings, and uses the fixed $g$ only in the source risk, which can be better estimated empirically.

If $\mathcal{F}$ is not too rich, the latent divergence can be empirically minimized by finding a well-aligned embedding. Hence, we can minimize the upper bound in Theorem 4 by minimizing the usual source loss and domain-invariant loss (2) and by choosing $\mathcal{F}$ and $\mathcal{G}$ appropriately to tradeoff the complexity penalty $d_{\mathcal{F}_{\mathcal{G}\Delta\mathcal{G}}}$, the latent divergence (which increases with complexity of $\mathcal{F}$ and decreases with complexity of $\mathcal{G}$), and the best in-class joint risk (which decreases with complexity of $\mathcal{F}$ and $\mathcal{G}$).

### 3.2 EMBEDDING COMPLEXITY TRADEOFFS EMPIRICALLY

To empirically verify the embedding complexity tradeoff, we keep the predictor class $\mathcal{F}$ fixed, vary the embedding class $\mathcal{G}$, and minimize the source loss and alignment objective (2). Concretely, we train domain adversarial neural networks (DANNs) (Ganin et al., 2016) on the Amazon reviews

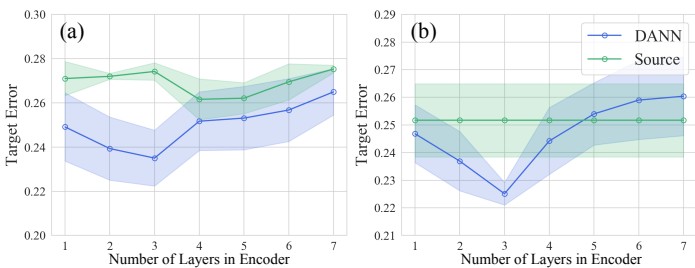

Figure 2: Empirical verification on Amazon reviews dataset. (a) Vary the number of layers in the encoder while fixing the predictor. (b) Fix the total number of layers and optimize the domain-invariant loss in different layers.

dataset (Book → Kitchen). Our hypothesis class is a multi-layer ReLU network, and the divergence is minimized against a discriminator. For more experimental details and results, please refer to section 6. We train different models by varying the number of layers in the encoder while fixing the predictor to $4$ layers. Figure 3.2(a) shows that, when increasing the number of layers in the encoder, the target error decreases initially and then increases as more layers are added. This supports our theory: the smaller encoders are not rich enough to allow for good alignments and $\lambda_{\mathcal{FG}}(g)$, but overly expressive encoders may overfit.

**Predictor Complexity.** Theoretically, the complexity of the predictor class $\mathcal{F}$ also affects the generalization bound in Theorem 4. Empirically, we found that the predictor complexity has much weaker influence on the target risk (see experiments in Appendix B). Indeed, theoretically, while the complexity of $\mathcal{F}$ affects the latent divergence, if the alignment via $g$ is very good, this divergence can still be small. In addition, the $\mathcal{F}_{\mathcal{G} \Delta \mathcal{G}}$-divergence is more sensitive to the embedding complexity than the predictor complexity. This offers a possible explanation for our observations. In the remainder of this paper, we focus on the role of the embedding.

**Discussion.** The results in this section indicate that, without constraining the embedding complexity, we may overfit the distribution alignment and thereby destroy label consistency as in Figure 1. The bound suggests to choose the minimal complexity encoder class $\mathcal{G}$ that is is still expressive enough to minimize the latent space divergence. Practically, this can be done by regularizing the encoder, e.g., restricting Lipschitz constants or norms of weight matrices. More explicitly, one may limit the number of layers of a neural network, or apply inductive biases via network architectures. For instance, compared to fully connected networks, convolutional neural networks (CNNs) restrict the output representations to be spatially consistent with respect to the input.

## 4 BOUNDS FOR MULTILAYER NEURAL NETWORKS

Due to their wide empirical success, multilayer neural networks have been adopted for learning domain-invariant representations. Next, we adapt the bound in Theorem 4 to multilayer networks. Specifically, we consider the number of layers as an explicit measurement of complexity. This will lead to a simple yet effective algorithm to mitigate the negative effect of very rich encoders.

### 4.1 EFFECT OF LAYER DIVISIONS

Assume we have an $N$-*layer feedforward neural network* $h \in \mathcal{H}$. The model $h$ can be decomposed as $h = f_i g_i \in \mathcal{F}_i \mathcal{G}_i = \mathcal{H}$ for $i \in \{1, 2, \ldots, N-1\}$ where the embedding $g_i$ is formed by the first layer to the $i$-th layer and the predictor $f_i$ is formed by the $i+1$-th layer to the last layer. We can then rewrite the bound in Theorem 4 in layer-specific form:

$$R_T(h) \leq R_S(h) + \underbrace{d_{\mathcal{F}_i \Delta \mathcal{F}_i}(p_S^{g_i}(Z), p_T^{g_i}(Z))}_{\text{Latent Divergence in } i\text{-th layer}} + \underbrace{d_{\mathcal{F}_i \mathcal{G}_i \Delta \mathcal{G}_i}(p_S, p_T)}_{\text{Embedding Complexity w.r.t } \mathcal{G}_i} + \lambda_{\mathcal{F}_i \mathcal{G}_i}(g_i). \quad (7)$$

This yields $N-1$ layer-specific upper bounds. Importantly, minimizing the domain-invariant loss in different layers leads to different tradeoffs between fit and complexity penalties. This is reflected by the following inequalities that relate different layer divisions.

**Proposition 5.** **(Monotonicity)** *In an $N$-layer feedforward neural network $h = f_i g_i \in \mathcal{F}_i \mathcal{G}_i = \mathcal{H}$ for $i \in \{1, 2, \ldots, N-1\}$, the following inequalities hold for all $i \leq j$:*

$$d_{\mathcal{F}_i \mathcal{G}_i \Delta \mathcal{G}_i}(p_S, p_T) \leq d_{\mathcal{F}_j \mathcal{G}_j \Delta \mathcal{G}_j}(p_S, p_T) \qquad \textit{(embedding complexity)} \quad (8)$$

$$d_{\mathcal{F}_i \Delta \mathcal{F}_i}(p_S^{g_i}(Z), p_T^{g_i}(Z)) \geq d_{\mathcal{F}_j \Delta \mathcal{F}_j}(p_S^{g_j}(Z), p_T^{g_j}(Z)) \qquad \textit{(latent divergence)} \quad (9)$$

Proposition 5 states that the latent divergence is monotonically decreasing and the complexity penalty is monotonically increasing with respect to the embedding's depth. This is a tradeoff within the fixed combined hypothesis class $\mathcal{H}$. A deeper embedding allows for better alignments and simultaneously reduces the depth (power) of $\mathcal{F}$; both reduce the latent divergence. At the same time, it incurs a larger $\mathcal{F}_{\mathcal{G}\triangle\mathcal{G}}$-divergence.

This suggests that there might be an optimal division that minimizes the bound on the target risk. In practice, this translates into the question: *in which intermediate layer should we optimize the domain-invariant loss?* Figure 3.2(b) shows how the target error changes as a function of the layer division, with a total of $n = 8$ layers. Indeed, empirically there is an optimal division with minimum target error, suggesting that for a fixed $\mathcal{H}$, i.e., total network depth, not all divisions are equal.

If the exact layer-specific bounds could be computed, one could simply select the layer division with the lowest bound. But, this is in general computationally nontrivial. Instead, we take a different perspective. In fact, the layer-specific bounds (7) all hold simultaneously, *independent of the layer we selected for distribution alignment.*

**Corollary 6.** *Let $h$ be an $N$-layer feedforward neural network $h = f_i g_i \in \mathcal{F}_i \mathcal{G}_i = \mathcal{H}$ for $i \in \{1, 2, \ldots, N - 1\}$, we have the layer-agnostic bound*

$$R_T(h) \leq R_S(h) + \min_{\{1 \leq i < N\}} \left\{ d_{\mathcal{F}_i \triangle \mathcal{F}_i}(p_S^{g_i}(Z), p_T^{g_i}(Z)) + d_{\mathcal{F}_{i\mathcal{G}_i \triangle \mathcal{G}_i}}(p_S, p_T) + \lambda_{\mathcal{F}_i \mathcal{G}_i}(g_i) \right\}.$$

*where $\lambda_{\mathcal{F}\mathcal{G}}(g)$ is the best in-class joint risk defined in Theorem 4.*

The corollary implies that at least one of these bounds should be small. Recall that the bounds depend on how well we can minimize the source risk and align the distributions via a sufficiently powerful embedding, while, at the same time, limiting the complexity of $\mathcal{F}$ and $\mathcal{G}$.

## 4.2 MULTILAYER DIVERGENCE MINIMIZATION AND REGULARIZATION

Corollary 6 points to various algorithmic ideas: (1) *Simultaneously* optimizing several bounds may result in approximately minimizing at least one of them, without having to select an optimal one. (2) We may attain small latent divergence with a deeper encoder, if we achieve to restrict the complexity of $\mathcal{G}$ appropriately. It turns out that these two ideas are related.

Optimizing the domain-invariant loss with alignment in a specific layer may result in large bounds for the other layers, due to the monotonicity of the two divergences (Proposition 5) and potentially non-aligned embeddings in lower layers. Hence, we propose to instead solve a multi-objective optimization problem where we *jointly* align source and target distributions in multiple layers. Let $\mathcal{L} \subseteq \{1, 2, \ldots, N - 1\}$ be a subset of layers. We minimize the weighted sum of divergences, and refer to this objective as *Multilayer Divergence Minimization (MDM)*:

$$\min_{h \in \mathcal{H}} \; R_S(h) + \sum_{i \in \mathcal{L}} \alpha_i d(p_S^{g_i}(Z), p_T^{g_i}(Z)). \tag{10}$$

This objective encourages alignment throughout the layer-wise embeddings in the network. First, a good alignment minimizes the latent divergence, if $\mathcal{F}$ is not too rich. For the lower layers (shallow embeddings), this comes together with a very restricted class of embeddings, and hence limits both latent divergence and complexity penalty. Without the optimization across layers, the embeddings in lower layers are not driven towards alignment.

Second, enforcing alignment in lower layers implicitly restricts the deeper embeddings in higher layers, since the embeddings are such that alignment happens early on. This effect may be viewed as an implicit regularization. By this perspective, the bounds for higher layers profit from low latent divergences (deeper embeddings and shallow predictors) and restricted complexity of $\mathcal{G}$.

In general, one can simply set $\mathcal{L} = \{1, 2, \ldots, N - 1\}$. To improve computational efficiency, we can sub-sample layers or exclude the first and the last few layers. MDM is simple and general, and can be combined with most algorithms for learning domain-invariant representations. For DANN, for instance, we minimize the divergence in multiple layers by adding discriminators.

## 5 OTHER RELATED WORKS

Existing approaches for learning domain-invariant representations may be distinguised, e.g., by which divergence they measure between source and target domain. Examples include domain adver-

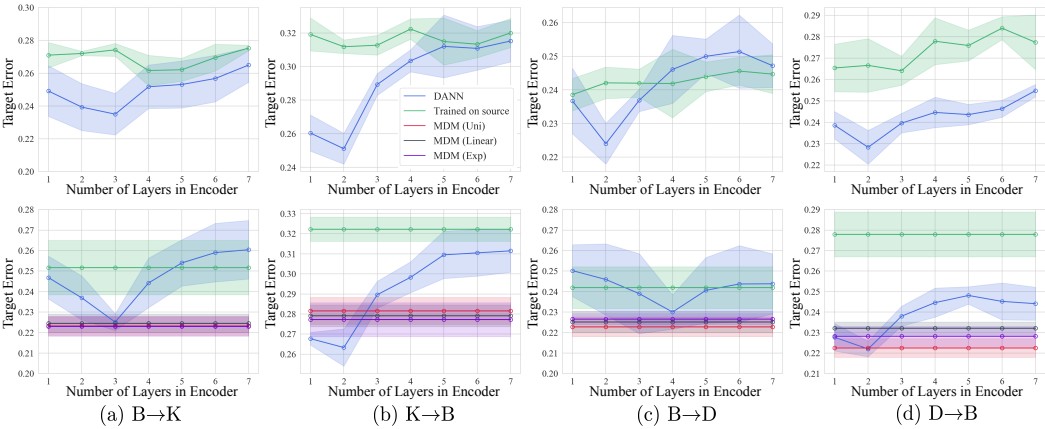

Figure 3: Amazon reviews dataset. First row: Fixed predictor class, varying number of layers in the encoder. Second row: Fixed total number of layers and optimizing domain-invariant loss in a single intermediate layer or MDM.

sarial learning approaches (Ganin & Lempitsky, 2014; Tzeng et al., 2015; Ganin et al., 2016), maximum mean discrepancy (MMD) (Long et al., 2014; 2015; 2016) and Wasserstein distance (Courty et al., 2016; 2017; Shen et al., 2017; Lee & Raginsky, 2018).

Other works improve performance by combining the domain-invariant loss with other objectives. Shu et al. (2018) penalize the violation of the cluster assumption. In addition to the shared feature encoder between source and target domain, Bousmalis et al. (2016) include private encoders for each domain to capture domain-specific information. Long et al. (2018) propose a domain discriminator that is conditioned on the cross-covariance of domain-specific embeddings and classifier predictions to leverage discriminative information. Besides the usual distribution alignment, Hoffman et al. (2017) further align the input space with a generative model that maps the target input distribution to the source distribution. These previous works can be interpreted as adding additional regularization via auxiliary objectives, and thereby potentially reducing the complexity penalty.

Some previous works also optimize the domain-invariant loss in multiple layers. Long et al. (2016) fuse the representations from a bottleneck layer and a classifier layer by a tensor product and minimize the domain divergence based on the aggregated representations. Joint adaptation networks (JADs) (Long et al., 2017) minimize the MMD in the last few layers to make the embeddings more transferable. MDM can be seen as a generalization of JADs that minimizes domain divergence in nearly every layer, driven by a strong theoretical motivation. Importantly, minimizing the divergence only in the last few layers could still be suboptimal, since the embeddings may not be sufficiently regularized.

## 6 EXPERIMENTS

We test our theory and algorithm on several standard benchmarks: sentiment analysis (Amazon reviews dataset), digit classification (MNIST, MNIST-M, SVHN) and general object classification (Office-31). In all experiments, we train DANN (Ganin et al., 2016), which measures the latent divergence via a domain discriminator (Jensen Shannon divergence). A validation set from the source domain is used as an early stopping criterion during learning. In all experiments, we use the Adam optimizer (Kingma & Ba, 2014) and a progressive training strategy for the discriminator (Ganin et al., 2016). We primarily consider three types of complexity: number of layers, number of hidden neurons, and inductive bias (CNNs). In all experiments, we retrain each model for 5 times and plot the mean and standard deviation of the target error.

For evaluating MDM, we consider three weighting schemes: uniform weights ($\alpha_i = \alpha_0$), linearly decreasing ($\alpha_i = \alpha_0 - c \times i$), and exponentially decreasing ($\alpha_i = \alpha_0 \exp(-c \times i)$) where $c \geq 0$. The decreasing weights encourage the network to minimize the latent divergence in the first few layers, where the embedding complexity is low. This may also further restrict the deeper embeddings. More experimental details can be found in Appendix C.

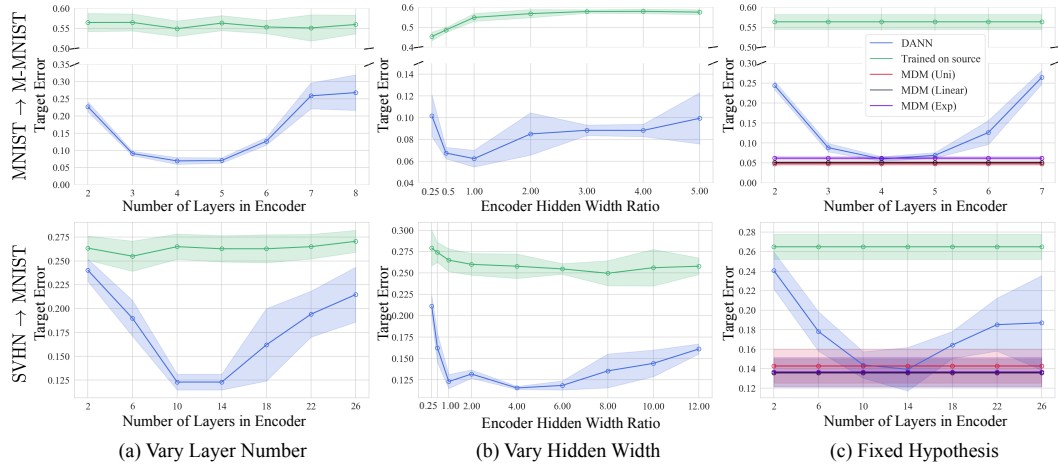

Figure 4: Digit classification. (a) Fixed predictor class, varying number of layers in the encoder. (b) Fixed predictor class, varying the hidden width of the encoder. (c) Fixed total number of layers and optimizing domain-invariant loss in a single intermediate layer or MDM.

**Sentiment Classification.** We first examine complexity tradeoffs on the Amazon reviews data, which has four domains (books (B), DVD disks (D), electronics (E), and kitchen appliances (K)) with binary labels (positive / negative review). Reviews are encoded into 5000 dimensional feature vectors of unigrams and bigrams. The hypothesis class are multi-layer ReLU networks. We show the results on B→K, K→B, B→D, and D→B in Figure 3. To probe the effect of embedding complexity by itself, we fix the predictor class to 4 layers and vary the number of layers of the embedding. In agreement with the results in Section 3.2, the target error decreases initially, and then increases as more layers are added to the encoder.

Next, we probe the tradeoff when the total number of layers is fixed to 8. The bottom row of Figure 3 shows that there exists an optimal setting for all tasks. For MDM, we optimize alignment in all intermediate layers. The results suggest that MDM's performance is comparable to the hypothesis with the optimal division, without tuning the division. The three weighting schemes perform similarly, suggesting that MDM is robust to weight selection.

**Digit Classification.** We next verify our findings on standard domain adaptation benchmarks: MNIST→MNIST-M (M→M-M) and SVHN→MNIST (S→M). We use standard CNNs as the hypothesis class; architecture details are in Appendix C.

To analyze the effect of the embedding complexity, we augment the original two-layer CNN encoders with 1 to 6 additional CNN layers for M→M-M and 1 to 24 for S→M, leaving other settings unchanged. Figure 4(a) shows the results. Again, the target error decreases initially and increase as the encoder becomes more complex. Notably, the target error increases by 19.8% in M→M-M and 8.8% in S→M compared to the optimal case, when more layers are added to the encoder. We also consider the width of hidden layers as a complexity measure, while fixing the depth of both encoder and predictor. The results are shown in Figure 4(b). This time, the decrease

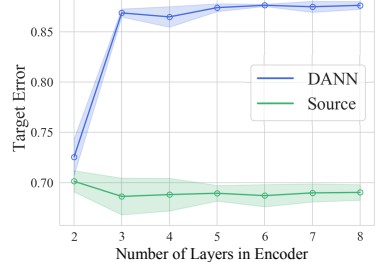

Figure 5: DANN with FC layers.

in target error is not significant compared to increasing encoder depth. This suggests that depth plays a more important role than width in learning domain-invariant representations.

Next, we fix the total number of CNN layers of the neural network to 7 and 26 for M→M-M and S→M, respectively, and optimize the domain-invariant loss in different intermediate layers. The results in Figure 4(c) again show a "U-curve", indicating the existence of an optimal division. Even with fixed total size of the network ($\mathcal{H}$), the performance gap between different divisions can still reach 19.5% in M→M-M and 10.4% in S→M. For MDM, $\mathcal{L}$ contains all the augmented CNN layers for M→M-M. For S→M, we sub-sample a CNN layer every four layers to form $\mathcal{L}$. We also observe

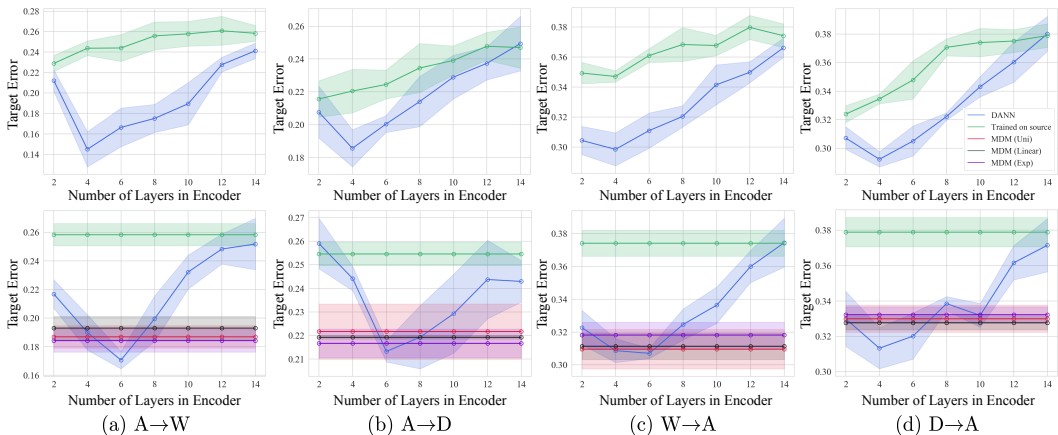

Figure 6: Office-31 Dataset. First row: Fixed predictor class, varying encoder depth. Second row: Fixed total number of layers, optimizing domain-invariant loss in a single layer or MDM.

that MDM with all weighting schemes consistently achieves comparable performance with the best division in S→M and even better performance in M→M-M.

To investigate the importance of inductive bias in domain-invariant representations, we replace the CNN encoder by an MLP encoder. The results for M→M-M are shown in Figure 5. Comparing to CNNs, which encode invariance via pooling and learned filters, MLPs do not have any inductive bias and lead to worse performance. In fact, the target error with MLP-based domain adaptation is higher than merely training on the source: without an appropriate inductive bias, learning domain invariant representations can even worsen the performance.

**Object Classification.** Office-31 (Saenko et al., 2010), one of the most widely used benchmarks in domain adaptation, contains three domains: Amazon (A), Webcam (W), and DSLR (D) with 4,652 images and 31 categories. We show results for A→W, A→D, W→A, and D→A in Figure 6. To overcome the lack of training data, similar to (Li et al., 2018; Long et al., 2018), we use ResNet-50 (He et al., 2016) pretrained on ImageNet (Deng et al., 2009) for feature extraction. With the extracted features, we adopt multi-layer ReLU networks as hypothesis class. Again, we increase the depth of the encoder while fixing the depth of the predictor to 2 and show the results Figure 6. Even with a powerful feature extractor, the embedding complexity tradeoff still exists. Second, we fix the total network depth to 14 and optimize MDM, with $\mathcal{L}$ containing all even layers in the network. MDM achieves comparable performance to the best division for most of the tasks, albeit slightly worse performance in D→A.

## 7 CONCLUSION

In this paper, we theoretically and empirically analyze the effect of embedding complexity on the target risk in domain-invariant representations. We find a complexity tradeoff that has mostly been overlooked by previous work. In fact, without carefully selecting and restricting the encoder class, learning domain invariant representations might even harm the performance. We further develop a simple yet effective algorithm to approximately optimize the tradeoff, achieving performance across tasks that matches the best network division, i.e., complexity tradeoff. Interesting future directions of work include other strategies for model selection, and a more refined analysis and exploitation of the effect of inductive bias.

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

## A  PROOFS

### A.1  PROOF OF THEOREM 4

**Theorem 4.** *For all $f \in \mathcal{F}$ and $g \in \mathcal{G}$,*

$$R_T(fg) \leq R_S(fg) + d_{\mathcal{F}\Delta\mathcal{F}}(p_S^g(Z), p_T^g(Z)) + d_{\mathcal{F}_\mathcal{G}\Delta\mathcal{G}}(p_S, p_T) + \lambda_{\mathcal{F}\mathcal{G}}(g).$$

*where $\lambda_{\mathcal{F}\mathcal{G}}(g)$ is the best in-class joint risk defined as*

$$\lambda_{\mathcal{F}\mathcal{G}}(g) = \inf_{f'\in\mathcal{F}, g'\in\mathcal{G}} 2R_S(f'g) + R_S(f'g') + R_T(f'g').$$

*Proof.* We first define the optimal composition hypothesis $f^*g^*$ with respect to an encoder $g$ to be the hypothesis which minimizes the following error

$$f^*g^* = \underset{f'\in\mathcal{F}, g'\in\mathcal{G}}{\arg\min} 2R_S(f'g) + R_S(f'g') + R_T(f'g') \tag{11}$$

By the triangle inequality for classification error (Ben-David et al. (2007)),

$$R_T(fg) \leq R_T(f^*g^*) + R_T(fg, f^*g^*) \tag{12}$$
$$\leq R_T(f^*g^*) + R_T(fg, f^*g) + R_T(f^*g, f^*g^*) \tag{13}$$

The second term in the R.H.S of Eq. 13 can be bounded as

$$R_T(fg, f^*g) \leq R_S(fg, f^*g) + |R_S(fg, f^*g) - R_T(fg, f^*g)| \tag{14}$$
$$\leq R_S(fg, f^*g) + \sup_{f,f'\in\mathcal{F}} |R_S(fg, f'g) - R_T(fg, f'g)| \tag{15}$$
$$= R_S(fg, f^*g) + d_{\mathcal{F}\Delta\mathcal{F}}(p_S^g(Z), p_T^g(Z)) \tag{16}$$
$$\leq R_S(fg) + R_S(f^*g) + d_{\mathcal{F}\Delta\mathcal{F}}(p_S^g(Z), p_T^g(Z)) \tag{17}$$

The third term in the R.H.S of Eq. 13 can be bounded as

$$R_T(f^*g, f^*g^*) \leq R_S(f^*g, f^*g^*) + |R_S(f^*g, f^*g^*) - R_T(f^*g, f^*g^*)| \tag{18}$$
$$\leq R_S(f^*g, f^*g^*) + \sup_{f\in\mathcal{F}, g,g'\in\mathcal{G}} |R_S(f'g, f'g') - R_T(f'g, f'g')| \tag{19}$$
$$= R_S(f^*g, f^*g^*) + d_{\mathcal{F}_\mathcal{G}\Delta\mathcal{G}}(p_S(X), p_T(X)) \tag{20}$$
$$\leq R_S(f^*g) + R_S(f^*g^*) + d_{\mathcal{F}_\mathcal{G}\Delta\mathcal{G}}(p_S(X), p_T(X)) \tag{21}$$

Combine the above bounds, we have

$$R_T(fg) \le R_S(fg) + d_{\mathcal{F}\Delta\mathcal{F}}(p_S^g(Z), p_T^g(Z)) + d_{\mathcal{F}_\mathcal{G}\Delta\mathcal{G}}(p_S(X), p_T(X)) + \lambda_{\mathcal{FG}}(g) \qquad (22)$$

where

$$\lambda_{\mathcal{FG}}(g) = 2R_S(f^*g) + R_S(f^*g^*) + R_T(f^*g^*) \qquad (23)$$

$$= \inf_{f' \in \mathcal{F}, g' \in \mathcal{G}} 2R_S(f'g) + R_S(f'g') + R_T(f'g') \qquad (24)$$

$$\square$$

## A.2   PROOF OF PROPOSITION 5

**Proposition 5.** *In an $N$-layer feedforward neural network $h = f_i g_i \in \mathcal{F}_i \mathcal{G}_i = \mathcal{H}$ for $i \in \{1, 2, \ldots, N-1\}$, the following inequalities hold for all $i \le j$:*

$$d_{\mathcal{F}_i \mathcal{G}_i \Delta \mathcal{G}_i}(p_S, p_T) \le d_{\mathcal{F}_j \mathcal{G}_j \Delta \mathcal{G}_j}(p_S, p_T)$$

$$d_{\mathcal{F}_i \Delta \mathcal{F}_i}(p_S^{g_i}(Z), p_T^{g_i}(Z)) \ge d_{\mathcal{F}_j \Delta \mathcal{F}_j}(p_S^{g_j}(Z), p_T^{g_j}(Z))$$

*Proof.* Given a class of multilayer feedforward neural network, We define a class of function $\mathcal{Q}_{ij}$ to represent the function class formed by the intermediate hidden layer $i$ to layer $j$.

We now prove the first inequality. By the definition of $\mathcal{F}_\mathcal{G}\Delta\mathcal{G}$-divergence, for every $i \le j$

$$d_{\mathcal{F}_i \mathcal{G}_i \Delta \mathcal{G}_i}(p_S, p_T) \qquad (25)$$

$$= \sup_{\substack{f \in \mathcal{F}_i \\ g, g' \in \mathcal{G}_i}} |R_S(fg, fg') - R_T(fg, fg')| \qquad (26)$$

$$= \sup_{\substack{f \in \mathcal{F}_j, q \in \mathcal{Q}_{ij} \\ g, g' \in \mathcal{G}_i}} |R_S(fqg, fqg') - R_T(fqg, fqg')| \qquad (27)$$

$$\le \sup_{\substack{f \in \mathcal{F}_j \\ q, q' \in \mathcal{Q}_{ij} \\ g, g' \in \mathcal{G}_i}} |R_S(fqg, fq'g') - R_T(fqg, fq'g')| \qquad (28)$$

$$= \sup_{\substack{f \in \mathcal{F}_j \\ g, g' \in \mathcal{G}_j}} |R_S(fg, fg') - R_T(fg, fg')| \qquad (29)$$

$$= d_{\mathcal{F}_j \mathcal{G}_j \Delta \mathcal{G}_j}(p_S, p_T) \qquad (30)$$

We next prove the second inequality. By the definition of $\mathcal{F}\Delta\mathcal{F}$-divergence, for every $i \le j$

$$d_{\mathcal{F}_j \Delta \mathcal{F}_j}(p_S^{g_j}(Z), p_T^{g_j}(Z)) \qquad (31)$$

$$= \sup_{f, f' \in \mathcal{F}_j} |R_S(fg_j, f'g_j) - R_T(fg_j, f'g_j)| \qquad (32)$$

$$= \sup_{f, f' \in \mathcal{F}_j} |R_S(fq_{ij}g_i, f'q_{ij}g_i) - R_T(fq_{ij}g_i, f'q_{ij}g_i)| \qquad (33)$$

$$\le \sup_{\substack{q \in \mathcal{Q}_{ij} \\ f, f' \in \mathcal{F}_j}} |R_S(fqg_i, f'qg_i) - R_T(fqg_i, f'qg_i)| \qquad (34)$$

$$\le \sup_{\substack{q, q' \in \mathcal{Q}_{ij} \\ f, f' \in \mathcal{F}_j}} |R_S(fqg_i, f'q'g_i) - R_T(fqg_i, f'q'g_i)| \qquad (35)$$

$$= \sup_{f, f' \in \mathcal{F}_i} |R_S(fg_i, f'g_i) - R_T(fg_i, f'g_i)| \qquad (36)$$

$$= d_{\mathcal{F}_i \Delta \mathcal{F}_i}(p_S^{g_i}(Z), p_T^{g_i}(Z)) \qquad (37)$$

$$\square$$

## B    Predictor Complexity

We investigate the effect of predictor complexity on MNIST→MNIST-M. Follow the procedure in section 6, we augment the original predictor with 1 to 7 additional CNN layers and fix the number of layers in encoder to 4 or vary the hidden width. The results are shown in Figure 7. The target error slightly decreases as the number of layers in the predictor increases. Even we augment 7 layers to the predictor, the target error only decrease $0.9\%$ which is nearly ignorable. Therefore, we focus on the embedding complexity in the main paper which is both theoretically and empirically interesting.

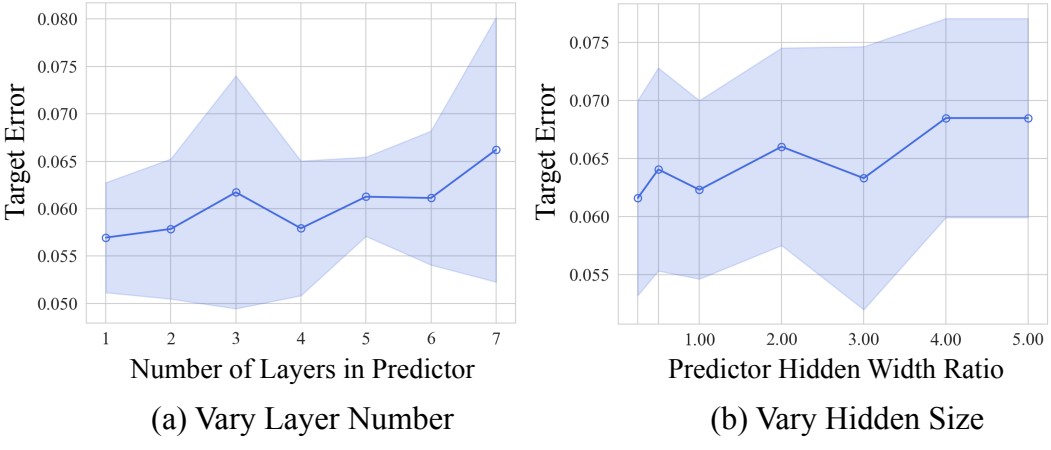

(a) Vary Layer Number           (b) Vary Hidden Size

Figure 7:   Predictor complexity trade-off on MNIST→MNIST-M. (a) Fix the encoder class and vary the number of layers in the predictor. (b) Fix the encoder class and vary the hidden width of the predictor.

## C    Experiment Details and Network Architectures

### C.1    Amazon Review Dataset

The learning rate of Adam optimizer is set to $1 \times e^{-3}$ and the model are trained for 50 epochs. We adopt the original progressive training strategy for discriminator (Ganin et al., 2016) where the weight $\alpha$ for domain-invariant loss in equation (2) is initiated at $0$ and is gradually changed to $1$ using the following schedule:

$$\alpha = \frac{2}{1 + \exp(-10 \cdot p)} - 1 \tag{38}$$

where $p$ is the training progress linearly changing from $0$ to $1$. The architecture of the hypothesis and discriminator are as follows:

| Encoder |
| --- |
| nn.Linear(5000, 128) |
| nn.ReLU |
| nn.Linear(128, 128) |
| nn.ReLU |
| $\times n$ (depends on the number of layers) |

| Predictor |
| --- |
| nn.Linear(128, 128) |
| nn.ReLU |
| $\times n$ (depends on the number of layers) |
| nn.Linear(128, 2) |
| nn.Softmax |

| Discriminator |
| --- |
| nn.Linear(128, 256) |
| nn.ReLU |
| nn.Linear(256, 256) |
| nn.ReLU |
| $\times 5$ |
| nn.Linear(256, 2) |
| nn.Softmax |

## C.2 DIGIT CLASSIFICATION

The learning rate of Adam optimizer is set to $1 \times e^{-3}$ and the model are trained for 100 epochs. The weight $\alpha$ for domain-invariant loss in equation (2) is initiated at 0 and is gradually changed to 0.1 using the same schedule in section C.1. The architecture of the hypothesis and discriminator are as follows:

| Encoder |
| :---: |
| nn.Conv2d(3, 64, kernel_size=5) |
| nn.BatchNorm2d |
| nn.MaxPool2d(2) |
| nn.ReLU |
| nn.Conv2d(64, 128, kernel_size=5) |
| nn.BatchNorm2d |
| nn.Dropout2d (only added for MNIST→MNIST-M) |
| nn.MaxPool2d(2) |
| nn.ReLU |
| nn.Conv2d(128, 128, kernel_size=3, padding=1) |
| nn.BatchNorm2d |
| nn.ReLU |
| $\times n$ (depends on the number of layers) |

| Predictor |
| :---: |
| nn.Conv2d(128, 128, kernel_size=3, padding=1) |
| nn.BatchNorm2d |
| nn.ReLU |
| $\times n$ (depends on the number of layers) |
| flatten |
| nn.Linear(2048, 256) |
| nn.BatchNorm1d |
| nn.ReLU |
| nn.Linear(256, 10) |
| nn.Softmax |

| Discriminator |
| :---: |
| nn.Conv2d(128, 256, kernel_size=3, padding=1) |
| nn.ReLU |
| nn.Conv2d(256, 256, kernel_size=3, padding=1) |
| nn.ReLU |
| $\times 4$ |
| Flatten |
| nn.Linear(4096, 512) |
| nn.ReLU |
| nn.Linear(512, 512) |
| nn.ReLU |
| nn.Linear(512, 2) |
| nn.Softmax |

In the hidden width experiments, we treat the architectures above as the pivot and multiply their hidden width with the ratios.

## C.3 OFFICE-31

We exploit the feature after average pooling layer of the ResNet-50 (He et al., 2016) pretrained on ImageNet (Deng et al., 2009) for feature extraction. The learning rate of Adam optimizer is set to $3 \times e^{-4}$ and the model are trained for 100 epochs. The weight $\alpha$ for domain-invariant loss in

equation (2) is initiated at $0$ and is gradually changed to $1$ using the same schedule in section C.1. The architecture of the hypothesis and discriminator are as follows:

| Encoder |
| --- |
| nn.Linear(2048, 256) |
| nn.ReLU |
| nn.Linear(256, 256) |
| nn.ReLU |
| $\times n$ (depends on the number of layers) |

| Predictor |
| --- |
| nn.Linear(256, 256) |
| nn.ReLU |
| $\times n$ (depends on the number of layers) |
| nn.Linear(256, 2) |
| nn.Softmax |

| Discriminator |
| --- |
| nn.Linear(256, 256) |
| nn.ReLU |
| $\times 6$ |
| nn.Linear(256, 2) |
| nn.Softmax |

## C.4 MULTILAYER DIVERGENCE MINIMIZATION

In all the experiments, we minimize the divergence in multiple layers by augmenting additional discriminators for each layer-specific representations where the discriminators share the same architecture as the standard setting.

For uniform weighting scheme ($\alpha_i = \alpha_0$), $\alpha_i$ is set to the normalized same value $\alpha$ in the stand setting. For linear decreasing scheme ($\alpha_i = \alpha_0 - c \times i$), $\alpha_i$ decreases from $\alpha_0 = \alpha$ to 0 linearly. For exponentially decreasing scheme ($\alpha_i = \alpha_0 \exp(-c \times i)$), $\alpha_0$ is set to $\alpha$ and $c$ increases from 0 to 2 linearly.

