# OpenReview forum: "The Role of Embedding Complexity in Domain-invariant Representations"
_ICLR.cc/2020/Conference — Reject_

### Official Review · AnonReviewer3 · 2019-10-24
**Official Blind Review #3**

**Rating:** 3

**Review:**

This paper proposes a new theory for domain adaptation considering the complexity of representation extractors. This paper gives a new bound for target error in domain adaptation, which contains the classic distribution distance related to the hypothesis space of high-level classifiers and a new distribution distance defined on the embedding space. This paper also proposes Multilayer Divergence Minimization algorithm based on the theory and evaluates it on real-world dataset.
Positive points:
(a) This paper proposes an interesting insight that the complexity of embeddings is also important in domain adaptation.
(b) This paper defines a new distribution divergence and build an interesting theory based on it.
(c) The proposed algorithm could automatically reach the best result of trying DANN on each layer.
Negative points:
(a) There is no proof that this new bound is better than classic domain adaptation theory (Ben-David et al., 2010). Although this bound involves new insight, the novelty is limited if it is looser than existing upper bound. Furthermore, there are no creative tools in the mathematical proof part, which is a direct extension of the classic theory.
(b) There is no analysis about the generalization when estimating this upper bound from finite samples. It could be easily seen that the sample complexity of embedding complexity is at least of the same order than classic \mathcal{H}\Delta\mathcal{H}-divergence (Ben-David et al., 2010).
(c) The analysis on the monotonicity of the divergences across the layers is very limited. It will be better if there is a discussion about when the monotonicity is strict.
(d) What is the role of embedding complexity in the algorithm? It seems that only high-level classifier divergence is minimized.
(e) Why minimizing the sum of divergences computed on all layers can control the proposed upper bound? It seems that if the embedding complexity of each layer is a constant, minimize divergence of a single layer can further minimize the minimum. Furthermore, there are previous method that minimizes divergences on all layers [A]. Please give a discussion on this method.
(f)The empirical evaluation is relatively weak. There is no experiment based on convolutional networks, which are widely used on the Digit and Office-31 datasets.

Although the insight is interesting, the novelty of this paper is not enough for being accepted by ICLR. So I vote for rejecting this submission.

[A] Zhang, Weichen, et al. "Collaborative and adversarial network for unsupervised domain adaptation." Proceedings of the IEEE Conference on Computer Vision and Pattern Recognition. 2018.


**Experience Assessment:**

I have published one or two papers in this area.

**Review Assessment: Checking Correctness Of Derivations And Theory:**

I carefully checked the derivations and theory.

**Review Assessment: Checking Correctness Of Experiments:**

I assessed the sensibility of the experiments.

**Review Assessment: Thoroughness In Paper Reading:**

I read the paper thoroughly.

---

> ### Author Response · Authors · 2019-11-11
> **Response to Reviewer #3**
>
> Thank you for your constructive comments. We would like to address your concerns as follows:
>
> (a) Our bound is tighter in some conditions. As we point out in definition 3,  theFG\DeltaG-divergence is smaller than the FG\DeltaFG-divergence. Therefore, comparing to (4), if the lambda in (4) and (6) are small enough and the latent divergence is sufficiently minimized, our bound can be smaller than the original bound from Ben-David.
>
> (b) Thank you for pointing this out. Quantifying the sample complexity of the upper bound is indeed an interesting question, and it would be interesting to add a discussion about it.
>
> (c) If the inequalities are strict, minimizing domain-invariant loss in different layers might achieves similar performance. Although we did not theoretically prove it, the experiments reveal that the monotonicity could be strict in practice.
>
> (d) The encoder is restricted in the sense that restricting the encoder to align the distributions in each layer implicitly restricts the set of feasible encodings. While we do not theoretically quantify this effect, we do validate the effect empirically, in the sense that we observe good empirical results. We will use careful wording and explain this in more detail.
>
> (e) As stated above, this approach implicitly restricts the feasible set of embeddings to those that align the distributions well in all layers. Empirically we find an effect on the performance that is well visible.
>
> Thank you for the pointer to [A]. We will cite and discuss this paper. As opposed to that work, our goal here is to find the most simple method to solve the layer-selection issue. Though MDM is simple, it resolves the layer selection problem we propose. We also provide a theoretical motivation.
>
> (f) We do use CNNs which is stated in the digit classification and object classification paragraph in section 6.
>
> Thank you again for your suggestions.
>
> Thanks,
> Authors

---

### Official Review · AnonReviewer2 · 2019-10-25
**Official Blind Review #2**

**Rating:** 1

**Review:**

This paper studies the problem of domain adaptation via learning invariant representations. The main argument here is that when the total depth of layers in a neural network is fixed, tradeoffs exist between feature alignment and prediction power. Furthermore, the authors argue that richer feature extractor can sometimes significantly overfit the source domain, leading to a large risk on the target domain.

Overall the paper is well-written and easy to follow. My major concern is that the paper, including the motivation and illustrative example, are too similar to previous work [1-2]. More detailed discussions are needed to highlight the difference of this work compared with [1-2]. The main contribution lies in Theorem 4. However, the upper bound is both loose and misleading. Compared with the original generalization upper bound [3], the one proposed in this paper contains a constant $\lambda$ that contains FOUR optimal error terms. Note that the original one in [3] only contains two such terms. In fact even a bound containing 2 such terms could potentially be very loose, since it's perfectly fine that a hypothesis can have large risk on the source domain while still attaining a small risk on the target domain. The bound is misleading in the sense that this $\lambda$ term cannot be computed or approximated, hence only the first two terms in (6) could be minimized in practice. However, this again can potentially lead to large target risk when the label distributions of source and target domains differ.

The experiments on using different number of layers of the network as feature extractors are quite interesting. The main message here is that general tradeoff exists with richer encoding function class. However, similar phenomenons have already been observed [4, Section 6.4], and it's not clear to me what's new here.

[1].    On Learning Invariant Representations for Domain Adaptation, ICML 2019.
[2].    Domain Adaptation with Asymmetrically-Relaxed Distribution Alignment, ICML 2019.
[3].    Analysis of representations for domain adaptation, NIPS 2007.
[4].    A DIRT-T APPROACH TO UNSUPERVISED DOMAIN ADAPTATION, ICLR 2018.

**Experience Assessment:**

I have published in this field for several years.

**Review Assessment: Checking Correctness Of Derivations And Theory:**

I carefully checked the derivations and theory.

**Review Assessment: Checking Correctness Of Experiments:**

I assessed the sensibility of the experiments.

**Review Assessment: Thoroughness In Paper Reading:**

I read the paper thoroughly.

---

> ### Author Response · Authors · 2019-11-11
> **Response to Reviewer #2**
>
> Thank you for your helpful suggestions. We would like to address your concerns as follows:
>
> 1. Comparison to Previous Works
>
> They and we are addressing a common issue, label consistency in domain-invariant representations, hence the seeming similarities. But all take a different perspective to focus on.
> In [1], they propose a lower bound on the target error. Again, they do not explicitly consider the embedding complexity. We point out that restricting the encoder is a necessary condition to achieve good performance in unsupervised domain adaptation.
> In [2], they provide an upper bound by leveraging the “connectedness” in input space and the Lipschitzness of the encoder. However, the input space distance between source and target domains has to be small to make the bound tighter, which might not hold in the common domain adaptation benchmarks (e.g. MNIST->MNIST-M).
>
> In comparison, the embedding complexity is not discussed in [1,2] and we believe that the insights from our theory and experiments are unique. With respect to the example, in contrast to [1-2], our example is motivated by the “embedding complexity” (main difference). For instance, in Figure 1, we compare two encoders with different complexity which is not shown in previous works.
>
>
> 2. About the upper bound
>
> In most of the domain adaptation bounds (including [1,2,3]), the goal is to use the source error to bound the target error. Therefore, without assuming the source error is small, not only our bound, the bounds in [1,2,3] are also large since the first term of the bound is the source error itself. As a consequence, assuming the additional two terms R_S(f^{\prime}g) are small is reasonable. Also, in most of the papers [1,2,3], the proposed bound can neither be computed or approximated.
>
> Comparing to [3], our bound is tighter if the source error and the latent divergence is sufficiently minimized. As we point out in Definition 3, the FG\DeltaG-divergence is smaller than the FG\DeltaFG-divergence. Therefore, comparing to (4) (the bound of [3]), if the lambda in (4) and (6) are small enough and the latent divergence is sufficiently minimized in (6), our bound (6) is smaller than the original bound (4) in [3] .
>
>
> 3. Layer-wise Tradeoff
>
> In [4, section 6.4], the performance is actually increasing along with the layer number and only decreases in the last layer. These results by themselves may be misleading for the questions we address, since it may look like increasing the number of layers will always improve performance. In our bounds and experiments, we observe that this is not the case, and increasing the layer number in the encoder can make the performance significant worse than the optimal case (Figure 4, (c)). As opposed to [4], we connect the results to theoretical results. We also include a different set of experiments. We will discuss this in more detail.
>
>
> Thank you again for your suggestions.
>
> Thanks,
> Authors

---

### Official Review · AnonReviewer4 · 2019-11-02
**Official Blind Review #4**

**Rating:** 3

**Review:**

This paper studies the impact of embedding complexity on domain-invariant representations. By incorporating embedding complexity into the previous upper bound explicitly, the authors demonstrate the limitations of previous theories and algorithms. Based on their theoretical findings, the authors propose to control the embedding complexity with implicit regularization. Specifically, aligning source and target feature distributions in multiple layers controls both embedding complexity and domain discrepancy. The proposed algorithm can achieve similar performance as DANN with manual selection of embedding depth.

By noting that the hypothesis space can be decomposed in to the feature extractor and the classifier, the authors propose to address the domain discrepancy separately. D_H\DeltaH is termed latent divergence, which the algorithm attempts to minimize. D_G\DeltaG is treated as embedding complexity, which is the intrinsic property of the feature extractor. Thus, domain-invariant representations should seek a proper tradeoff between those two terms.

The paper is well-written and the contributions are stated clearly. The exploration on the layer division is really insightful.

However, I have several concerns:
1.	The proposed upper bound is insightful, but it has several limitations. Compared to the version applied to the feature space in equation (3), the proposed upper bound is looser. The embedding complexity terms includes two encoders, which are deep neural networks in practice, thus it can be excessively large. As the authors point out, in equation (3), the embedding complexity is not addressed explicitly, but it is implicit in the adaptability \lambda in a more reasonable way. Previous works [1], [2], [3] have already taken them into consideration. Proposition 5 is a direct application of proposition 1 in [1].
2.	On the claim of implicit regularization. By applying domain adversarial training to multiple layers, the authors claim that the encoder in higher layers is implicitly restricted. However, they do not validate this regularization effect. Is the embedding complexity controlled? Theoretical analysis or experimental results would be helpful.
3.	The proposed MDM method seems to be incremental. [4] has probed into the effect of multi-layer adaptation strategy. Besides, applying domain adversarial training to many layers leads to more computational cost and may slow down training significantly.


[1]Fredrik D Johansson, Rajesh Ranganath, and David Sontag. Support and invertibility in domain- invariant representations. arXiv preprint arXiv:1903.03448, 2019.
[2]Han Zhao, Remi Tachet des Combes, Kun Zhang, and Geoffrey J Gordon. On learning invariant representation for domain adaptation. arXiv preprint arXiv:1901.09453, 2019.
[3] Hong Liu, Mingsheng Long, Jianmin Wang, and Michael Jordan. Transferable adversarial training: A general approach to adapting deep classifiers. In International Conference on Machine Learning, pp. 4013–4022, 2019.
[4] Mingsheng Long, Yue Cao, Jianmin Wang, and Michael I. Jordan. Learning transferable features with deep adaptation networks. In Proceedings of the 32nd International Conference on International Conference on Machine Learning, volume 37, pp. 97–105, 2015.


**Experience Assessment:**

I have published one or two papers in this area.

**Review Assessment: Checking Correctness Of Derivations And Theory:**

I carefully checked the derivations and theory.

**Review Assessment: Checking Correctness Of Experiments:**

I assessed the sensibility of the experiments.

**Review Assessment: Thoroughness In Paper Reading:**

I read the paper at least twice and used my best judgement in assessing the paper.

---

> ### Author Response · Authors · 2019-11-11
> **Response to Reviewer #4**
>
> Thank you for your constructive comments. Below we would like to address your concerns.
>
> 1. The upper bound
>
> Our bound is not looser. As we point out in the paper, the lambda in equation (3) can be arbitrarily large when the label is not consistent. It is not clear to us that why eqn (3) is “more reasonable”? We propose a term that includes the embedding complexity explicitly. We admit that complexity bounds for neural networks can be high, theoretically. Yet, our experiments show that the trends indicated in our bounds indeed appear to occur in practice.
>
> Comparing to [1][2][3], we analyze the problem from the perspective of embedding complexity.
> The bound in [1] does not explain the effect of the encoder, while our bound expresses it explicitly. In addition, [1] relies on common support, which does not hold for popular domain adaptation benchmarks (MNIST -> SVHN, Office31). We do not need such assumptions.
> In [2], they propose a lower bound on the target error. Again, they do not explicitly consider the embedding complexity. We point out that restricting the encoder is a necessary condition to achieve good performance in unsupervised domain adaptation.
> In [3], they address the label consistent problem with generates examples to fill in the gap between the source and target domains.
> It seems that in [1,2,3], the embedding complexity is not discussed and we believe that the insights from our theory and experiments are unique. We will add more discussion of related work to the paper. With regard to proposition 5, we admit that it is a generalization of it, and we will add that to the paper. However, we use the result to explain the layer-wise trade-off, which is different from [1].
>
>
> 2. Implicit regularization
>
> The encoder is restricted in the sense that it has to learn aligned embeddings in all layers,  instead of being free to choose arbitrary ones. While we do not theoretically quantify this effect, we do validate the effect empirically, in the sense that we observe good empirical results. We will use careful wording and explain this in more detail.
>
>
> 3. The proposed method
>
> Indeed, similar approaches can be seen in other papers. However, our goal here is to find the simplest method to solve the layer-selection issue. Though MDM is simple, it resolves this problem, with a theoretical motivation. We will add the references and discuss related work in greater detail.
>
> Thank you again for your suggestions.
>
> Thanks,
> Authors

---

### Decision · Program_Chairs · 2019-12-19

**Decision:**

Reject

**Comment:**

This paper studies the impact of embedding complexity on domain-invariant representations by incorporating embedding complexity into the previous upper bound explicitly.

The idea of embedding complexity is interesting, the exploration has some useful insight, and the paper is well-written. However, Reviewers and AC generally agree that the current version can be significantly improved in several ways:
- The proposed upper bound has several limitations such as looser than existing ones.
- The embedding complexity is only addressed implicitly, which shares similar idea with previous works.
- The claim of implicit regularization has not been explored in-depth.
- The proposed MDM method seems to be incremental and related closely with the embedding complexity.
- There is no analysis about the generalization when estimating this upper bound from finite samples.

There are important details requiring further elaboration. So I recommend rejection.